# Improved Therapeutic Efficacy of CBD with Good Tolerance in the Treatment of Breast Cancer through Nanoencapsulation and in Combination with 20(S)-Protopanaxadiol (PPD)

**DOI:** 10.3390/pharmaceutics14081533

**Published:** 2022-07-22

**Authors:** Jingxin Fu, Kunfeng Zhang, Likang Lu, Manzhen Li, Meihua Han, Yifei Guo, Xiangtao Wang

**Affiliations:** 1Institute of Medicinal Plant Development, Chinese Academy of Medical Sciences & Peking Union Medical College, No.151, Malianwa North Road, Haidian District, Beijing 100193, China; fujingxin@implad.ac.cn (J.F.); lulikang@implad.ac.cn (L.L.); limanzhen@implad.ac.cn (M.L.); mhhan@implad.ac.cn (M.H.); yfguo@implad.ac.cn (Y.G.); 2School of Pharmacy, Henan University of Chinese Medicine, Zhengzhou 450046, China; zkfgcx@163.com

**Keywords:** Cannabidiol (CBD), 20(S)-Protopanaxadiol (PPD), liposomes, breast cancer, anti-tumor efficacy

## Abstract

Cannabidiol (CBD), a nonpsychoactive major component derived from *Cannabis sativa*, widely used in neurodegenerative diseases, has now been proven to have growth inhibitory effects on many tumor cell lines, including breast tumors. Meanwhile CBD can effectively alleviate cancer-associated pain, anxiety, and depression, especially tumor cachexia, thus it is very promising as an anti-tumor drug with unique advantages. 20(S)-Protopanaxadiol (PPD) derived from the best-known tonic Chinese herbal medicine Ginseng was designed to be co-loaded with CBD into liposomes to examine their synergistic tumor-inhibitory effect. The CBD-PPD co-loading liposomes (CP-liposomes) presented a mean particle size of 138.8 nm. Further glycosyl-modified CP-liposomes (GMCP-liposomes) were prepared by the incorporation of n-Dodecyl β-D-maltoside (Mal) into the liposomal bilayer with glucose residue anchored on the surface to act as a ligand targeting the GLUT1 receptor highly expressed on tumor cells. In vivo studies on murine breast tumor (4T1 cells)-bearing BALB/c mice demonstrated good dose dependent anti-tumor efficacy of CP-liposomes. A high tumor inhibition rate (TIR) of 82.2% was achieved with good tolerance. However, glycosylation modification failed to significantly enhance TIR of CP-liposomes. In summary, combined therapy with PPD proved to be a promising strategy for CBD to be developed into a novel antitumor drug, with characteristics of effectiveness, good tolerance, and the potential to overcome tumor cachexia.

## 1. Introduction

Cancer is one of the major public health problems throughout the world and leads to a large number of deaths each year. For women, the three most common cancers are breast, lung, and colorectal, accounting for 50% of all new diagnoses; breast cancer alone accounts for 30% of female cancers [1]. As the most lethal cancer among women globally, breast cancer has attracted much attention worldwide and many conventional therapies have been developed based on surgery, chemotherapy and/or radiotherapy, among which some have become the gold standard for its treatment in the past few decades. However, their therapeutic efficacy is often restricted by serious side effects and tumor resistance [2,3,4]. Therefore, a less invasive yet effective and tolerant cancer therapy is in great medical need. For this purpose, cannabidiol (CBD) therapy may be a promising strategy [5].

As a traditional herbal medicine, *Cannabis sativa* L. and its extracts have been used in the treatment of many diseases since 500 BC in Asia, such as glaucoma, anxiety, nausea, depression, neuralgia and so on [6,7]. More than 560 compounds have been separated from cannabis, most with specific biological and chemical activities [8], among which delta-9-tetrahydrocannabinol (Δ^9^-THC), cannabidiol (CBD), cannabichromene (CBC) and cannabigerol (CBG) are the most well-known [9,10]. Δ^9^-THC (Figure 1b) is the main component and is well known for its psychoactive effects, leading to the addiction of Marijuana.

As the second major component, CBD (Figure 1a) is not associated with psychoactivity [9,11] and has therefore been applied in the treatment of a wide range of neurodegenerative diseases [12,13] and tumors. CBD has shown growth inhibitory effects on glioblastoma [14], leukemia [15], lung cancer [16], breast cancer [17], cervical cancer [18], prostate cancer [19], and melanoma [20]. Various breast cancer cell lines, including estrogen-receptor (ER)-positive or negative or triple-negative breast cancer (TNBC) cells, have displayed a dose-dependent response to CBD (mostly with low IC50 values [17,21,22,23,24,25,26]).

CBD inhibits breast cancer cell growth via a variety of mechanisms, including apoptosis, autophagy, cell cycle arrest and gene expression regulation [17,21,23]. CBD could induce apoptosis of MDA-MB-231 cells involving caspase-3 and arrest the cell cycle of MCF-7, MDA-MB-231 and 4T1 cells at the G1/S checkpoint [21]. CBD downregulates mTOR, AKT, 4EBP1, and cyclin D while upregulating the expression of PPAR and its nuclear localization to exert its pro-apoptotic effect and induces autophagy and apoptosis [17,23]. CBD inhibits the activation of the EGF/EGFR signaling pathway and its downstream targets such as AKT, ERK and NF-kb, to inhibit the 4T1-mediated breast cancer cell line [22,27].CBD can also regulate the tumor microenvironment [28] and inhibit the migration, invasion and metastasis of aggressive breast cancer in vitro and in vivo [21,22,26,29].

It is reported that CBD can affect many tumoral features and molecular pathways [30]. Much of CBD’s anti-tumor activity is via its regulation of reactive oxygen species (ROS), modulating the tumor microenvironment, and immune modulation [31]. CBD causes the early production of ROS, depletion of intracellular glutathione, triggering caspase activation and oxidative stress in human glioma cells [32], leukemia cells [33] and breast cancer cells [23]. CBD can also lead to a loss of mitochondrial membrane potential in human lung cancer cells [16] and cervical cancer cells [34].

CBD could potently inhibit exosomes and microvesicles (EMV) release from many cancer cell lines, which can sensitize cancer cells to chemotherapeutic agents and reduce cancer growth in vivo [35]. CBD treatment causes the dose-related down-regulation of the ERK and Akt prosurvival signaling pathways and decreased hypoxia inducible factor HIF-1a expression in U87-MG cells [30]. CBD can inhibit the expression of GPR55, which is related directly or indirectly with changes that promote malignant growth, including uncontrolled cancer cell proliferation, angiogenesis, cancer cell adhesion, cancer cell migration, and metastasis [36].

Another unique advantage of CBD as an anti-tumor agent is that, CBD, alone or sometimes in combination with THC, can effectively alleviate cancer-associated pain, anxiety and depression, sleep problems, nausea and vomiting, oral mucositis and especially tumor cachexia [37].

However, as a highly lipophilic compound, CBD has poor water solubility (about 12.6 µg/L) and is unstable in gastric pH, highly susceptible to first-pass metabolism [38]. In the FDA approved oral drug (Epidiolex^®^) in 2018, sesame oil and alcohols were used as solubilizing agents to produce a purified CBD solution for the treatment of childhood epilepsy [38]. The formulation research of CBD reported so far includes PLGA and PCL microparticles [39,40], self-nanoemulsions [41], dihydroartemisinin-conjugate [42], polymeric micelles [43], lipid nanoparticles [44] and transfersomes [45]. However, most were designed to improve oral bioavailability, are unsuitable for intravenous administration or have achieved limited in vivo antitumor efficacy.

It is reported that CBD is a safe compound due to its low toxicity, high tolerability, and lack of side effects, even after chronic administration of high doses [9,46,47]. This also means that CBD alone may only lead to limited in vivo antitumor efficacy, which has been evidenced in an U87MG tumor-bearing mice model. In the preliminary study, various nanosuspensions and polymeric micelles were tried using many pharmaceutic adjuvants such as Poloxamer 188, D-α-tocopherol acid polyethylene glycol succinate (TPGS), DSPE-mPEG2000, mPEG2000-PCL2000, bovine serum albumin, sodium oleate, and PLGA-COOH as a stabilizer or carrier, aiming at a mean particle size of <200 nm, drug-loading >10%, being stably stored at room temperature for at least 7 days and being suitable for intravenous administration (limited particle size enlargement in normal saline or 5% glucose and in plasma). Unfortunately, no formulation met these requirements until liposomes were tried. While liposomes and liposomal drugs tend to accumulate in the liver after intravenous administration, this may increase hepatic injury or hepatotoxicity and result in safety issues, such as when the encapsulated drug itself had hepatotoxicity, as did many chemotherapeutics. Fortunately, as an FDA-approved medicine, CBD has very good tolerance; so far, there has been no report about significant hepatotoxicity or obvious liver injury for CBD [48,49]. Meanwhile, PPD shows no hepatoxicity either [50]. In order to obtain an effective but well-tolerated CBD-based antitumor formulation, in this paper, 20(S)-Protopanaxadiol (PPD, Figure 1d) is designed to be co-encapsulated with CBD in liposomes to examine their synergistic antitumor efficacy. 20(S)-Protopanaxadiol (PPD), as the common aglycone of ginsenoside Rg3 and Rh2, showed more potent in vitro and in vivo antitumor activity [51,52] than Rg3 and Rh2 [53]. PPD has a similar structure to that of cholesterol (Figure 1c) and novel liposomes prepared through the replacement of cholesterol by PPD showed very high drug-loading content for PPD [54]. Ginsenoside Rg3 and Rh2 (Figure 1f,e) were also used as a substitute for cholesterol to prepare liposomes with special functions, including long circulation and tumor targetability, which was claimed to be mostly due to the carbohydrate residues left on the surface of the liposomes [55]. Therefore, n-Dodecyl β-D-maltoside (Mal) was incorporated into the liposomal bilayer with two glucosyl residues on the surface to mimic the long circulating and tumor-targeting function depicted in the article [22].

## 2. Materials and Methods

### 2.1. Materials

CBD was provided by Yunnan Hansu Biochemical Co. Ltd. (Kunming, China). n-Dodecyl β-D-maltoside was from Shanghai Maclin Biochemical Co. Ltd. (Shanghai, China). Cholesterol was from Shanghai Yuanye Biotechnology Co., LTD (Shanghai, China). PPD was bought from Nanjing Spring Autumn Biological Engineering Co. Ltd. (Nanjing, China). DiR iodide [1-1-dioctadecyl-3,3,3,3-tetramethlindotricarboc-yanine iodide] (DiR) was purchased from AAT BioQuest (Sunnyvale, CA, USA). Paclitaxel (PTX) injection was obtained from Beijing Union Pharmaceutical Factory (Beijing, China). 3-(4,5-dimethylthiazol-2-yl)-2,5-diphenyltetrazolium bromide (MTT) were provided by Sigma-Aldrich (St. Louis, MO, USA). Acetonitrile (HPLC grade) was from Fisher Scientific (Pittsburgh, PA, USA). Deionized water was used during the experiments.

### 2.2. Animals and Cell Lines

The 4T1 (breast cancer) cell line was supplied by the national Infrastructure of Cell Line Resource (Beijing, China). The cells were cultured in Roswell Park Memorial Institute 1640 medium (RPMI 1640, Hy Clone) with 10% content of fetal bovine serum (FBS, Gibco, New York, NY, USA), 100 U/mL streptomycin (Gibco, New York, NY, USA), 100 U/mL penicillin (Gibco, New York, NY, USA) with 5% CO_2_ at 37 °C. Female BALB/c mice (20 ± 2 g, 6–8 weeks old) were obtained from SPF (Beijing) Biotechnology Co. Ltd. (SPF grade). The experimental animals were provided ad libitum feeding and were adapted to the environment of SPF-class housing in the laboratory for 7 days before experimentation.

### 2.3. Preparation of CBD-PPD-Liposomes (CP-Liposomes) and Glucose Modified CBD-PPD-Liposomes (GMCP-Liposomes)

CBD-PPD-liposomes (CP-liposomes) were prepared using film-rehydration combined with the probe ultrasonication method (Figure 2). Briefly, Soybean phosphatidylcholine (SPC), PPD (as the membrane stabilizer instead of cholesterol) and CBD with the mass ratio of 4:1:1 was co-solubilized in the anhydrous ethanol in a round-bottom flask. The organic solvent was removed by vacuum rotary evaporation at 40 °C until a uniform thin film formed on the inner surface of the flask bottom. Then, deionized water was added into the flask, stirred for 20 min in a 60 °C water bath and then sonicated using an ultrasonic probe (325 w, 20 min) to obtain the CP-liposomes.

Glycoside modified CBD-PPD-liposomes (GMCP-liposomes) were prepared as the above-mentioned procedure, except that n-Dodecyl β-D-maltoside (Mal) was added in the formulation and co-dissolved in anhydrous ethanol together with SPC, PPD and CBD with the mass ratio of SPC: PPD: CBD: Mal being 4:1:1:1.

DiR, a lipophilic, near-infrared fluorescent anthocyanin dye, is often incorporated into liposomes and nanoparticles to trace their in vivo biodistribution by dynamic imaging. DiR labeled CP-liposomes and GMCP-liposomes were also prepared as the procedure mentioned above, except that DiR was added in the ethanol solution with the mass ratio of CBD: DiR being 40:1.

### 2.4. Particle Size Distribution and Zeta Potential Measurement

The particle size distribution and zeta potential of CP-liposomes and GMCP-liposomes were determined by dynamic light scattering (DLS) with a Zetasizer Nano ZS system (Malvern Instruments Ltd., Malvern, UK) at 25 °C, which integrated phase analysis light scattering (λ = 633 nm) and noninvasive backscatter optics (scattering angle θ = 173°). Each sample was measured in triplicate and all data were expressed as the mean ± standard deviation (SD).

### 2.5. Morphology of CP-Liposomes

The morphology of CP-liposomes and GMCP-liposomes were observed by a JEM-1400 transmission electron microscope (JEOL, Tokyo, Japan) using a negative stain method. Briefly, one drop of sample solution was added on the 300-mesh copper grid, dried at room temperature, then dyed with uranyl acetate for 90 s. Then, the morphology was observed under TEM at an accelerating voltage of 120 kV.

### 2.6. Stability of CP-Liposomes and GMCP-Liposomes in Vairous Physiological Media

The CP-liposomes and GMCP-liposomes were incubated with the same volume of 1.8% NaCl, 10% glucose, 2 × PBS (pH 7.4), plasma or four times the volume of the simulated gastric fluid (1% pepsin in 1 mol/L diluted HCl) and simulated intestinal fluid (1% pancreatin in pH 6.8 PBS, 0.01 M) (1:4, *v*/*v*) at 37 °C. The size and particle size distribution were monitored by DLS at specific time intervals. Each sample was performed in triplicate.

### 2.7. HPLC Analysis

The CBD concentration in liposomes was measured using the HPLC system (DIONEX Ultimate 3000, Sunnyvale, CA, USA). A Symmetry C18 column (250 mm × 4.6 mm, 5 μm, Venusil) was used at 25 °C for chromatographic separation. The mobile phase constituted 0.1% acetate acetonitrile and 0.1% acetate water (77:32, *v*/*v*) at a flow rate of 0.8 mL/min. The detection wavelength UV was 220 nm.

### 2.8. Drug Loading Content

To determine the drug loading content (*DLC*) of CP-liposomes and GMCP-liposomes, the lyophilized CP-liposome powder was weighed and fully dissolved in methanol. The concentration of CBD was determined by HPLC analysis. *DLC* was calculated by the following formula,
$$DLC\left(\%\right)=\frac{c\times V}{W}\times 100\%$$
(*c*: concentration of CBD, *V*: volume of methanol solution of lyophilized CP-liposome powder, *W*: weight of lyophilized powder of CP-liposomes)

### 2.9. Differential Scanning Calorimetry (DSC) Characterization

The DSC thermal profile was obtained using a differential scanning calorimeter (DSC Q200 V24.4 Build 116). A suitable amount of SPC, PPD, CBD, Mal, lyophilized CP-liposome powder, lyophilized GMCP-liposome powder, the physical mixture (SPC, PPD, CBD) according to the formulation of CP-liposome and the physical mixture (SPC, PPD, CBD, Mal) according to the formulation of GMCP-liposome was sealed in a standard aluminum pan and detected from 0 to 350 °C (10 °C/min, a nitrogen atmosphere).

### 2.10. X-ray Diffraction (XRD) Measurements

X-ray diffraction (XRD) measurements of powder sample (SPC, PPD, CBD, Mal, lyophilized CP-liposome powder, lyophilized GMCP-liposome powder, the physical mixture (SPC, PPD, CBD) according to the formulation of CP-liposome, the physical mixture (SPC, PPD, CBD, Mal) according to the formulation of GMCP-liposome) was performed using an X-ray diffractometer (DX-2700, Dandong, China) with Cu-Kα radiation generated at 100 mA and 40 kV. Samples were scanned over an angular range of 3–80° of 2θ, with a step size of 0.02° and a count time of 3 s per step. Samples were kept rotating at 30 rpm during the analysis.

### 2.11. In Vitro Drug Release

The release of CBD from CP-liposomes and GMCP- liposomes in PBS (0.01 M, pH 7.2–7.4 containing 0.5% *w*/*v* tween 80) were conducted using the dialysis method (molecular weight cut-off 8–10 kDa) in a 37 °C thermostatic water bath under continuous stirring. At the predetermined time intervals, one milliliter of the external liquid was collected for HPLC analysis and replenished with an equivalent amount of fresh release medium. The dissolution medium was renewed every 24 h. The CBD concentration in the dialysate were determined by HPLC and the cumulative release rate was calculated and profiled. The above experiments were performed in triplicate.

### 2.12. MTT Assays

It was reported that cannabidiol was sensitive to multiple tumor cell lines [23,56,57]. MTT assay was carried out to examine the in vitro cytotoxicity of cannabidiol, CP-liposomes and GMCP-liposomes against the 4T1 murine breast cancer cell line. 4T1 cells (2500 cells/well in 200 μL) were seeded in 96-well plates and incubated at 37 °C and in 5% CO_2_ atmosphere for 24 h. Different CBD equivalent concentration of CP-liposomes and GMCP-liposomes (diluted in RPMI-1640 medium) were added (150 μL per well) and incubated for 72 h, using free CBD as a control and RPMI-1640 medium as a negative control. Then, 20 μL of MTT solution (5 mg/mL) was added to each well and incubated for another 4 h. Then, the medium was removed and 150 μL of DMSO was added to dissolve the formazan crystals with full vibration. The absorbance value of the supernatant in each well was measured at 570 nm using the ELISA plate reader. The cell viability rate was calculated according to the following formula.
$$\mathrm{Cell}\;\mathrm{viability}\;\mathrm{rate}\left(\%\right)=\left(1-\frac{OD_{t}}{OD_{n}}\right)*100\%$$
(where *OD_t_* means the absorbance value of the test groups, *OD_n_* means the absorbance value of the negative control groups)

The IC_50_ value of each group was calculated through GraphPad Prism software, Version 7 (GraphPad Software, Inc., San Diego, CA, USA). The MTT assay of other tumor cell lines (MCF-7, A549, Hela, etc.) were operated as the same procedure.

### 2.13. In Vivo Antitumor Efficacy

In the limited reports on in vivo antitumor studies, CBD alone achieved less than 20% in TIR against the triple-negative breast cancer at 10 mg/kg [58] and about 43% in TIR against U87MG tumor xenografts at 15 mg/kg [59]. Thus, we selected 15 mg/kg of liposomal CBD as a medium dose, 5 mg/kg of liposomal CBD as a low dose, and 45 mg/kg of liposomal CBD as a high dose to assess the dose-dependent antitumor efficacy of CP-liposomes. Since CP-liposomes contain 15 mg/kg of CBD and 15 mg/kg of PPD, 15 mg/kg of PPD-liposomes and 15 mg/kg of CBD-liposomes were intravenously administrated as a control to see if the combined CBD+PPD therapy could achieve synergistic therapeutic efficacy. GMCP-liposomes (15 mg/kg) were also examined with the aim to see if the surface glycosylation could further improve the in vivo antitumor efficacy of CP-liposome via GLUT1 mediated endocytosis.

Female BALB/c mice weighed 20 ± 2 g bearing 4T1 tumors were selected for the in vivo antitumor efficacy investigation. As shown in Appendix A, 0.2 mL 4T1 cell suspension (1 × 10^6^ cells) were injected in the right armpit of the mice subcutaneously. When the tumor volume reached to 100 mm^3^, the 4T1 tumor-bearing mice were randomly divided into 9 groups (9 mice in each group). The negative control group was intravenously injected with 0.2 mL of normal saline, while the positive control group was administrated with 0.2 mL of PTX injection (8 mg/kg). Three test groups were intravenously injected with 5, 15 and 45 mg/kg of CP-liposomes, respectively (according to the equivalent CBD dose). Another group was dosed with 15 mg/kg of GMCP-liposomes (the equivalent CBD dose). PPD liposomes (15 mg/kg of PPD) and CBD liposomes (15 mg/kg of CBD) were also dosed as a control to evaluate the in vivo synergistic effect of PPD and CBD. All the above groups were intravenously administrated through the tail vein every 2 days, 7 times. The last group was orally administrated with CP-liposomes (45 mg/kg of CBD) every 2 days as an oral control. The tumor volume and body weight of each mouse were measured during the whole experimental process. The tumor volume was calculated by the formula V = (a ∗ b^2^)/2.

At the end of the experiment, the mice were sacrificed unless otherwise specified, and the tumors, livers, and spleens were dissected and weighed to calculate the tumor inhibition rate (*TIR*%), liver index rate (*LIR*%) and spleen index rate (*SIR*%) according to the following formulas,
$$TIR\left(\%\right)=\left(1-\frac{W_{t}}{W_{n}}\right)\times 100\%$$

(*W_t_* is the mean tumor weight of mice in test groups and *W_n_* is the mean tumor weight of mice in the negative control group)
$$LIR=\overset{9}{\underset{i=1}{\sum }}\begin{array}{c} \frac{W_{l}}{W_{b}}÷9 \end{array}$$

(*W_l_* is the liver weight and *W_b_* is the body weight)
$$SIR=\overset{9}{\underset{i=1}{\sum }}\begin{array}{c} \frac{W_{s}}{W_{b}}÷9 \end{array}$$

(*W_s_* is the spleen weight and *W_b_* is the body weight)

### 2.14. The In Vivo Biodistribution of CP-Liposomes and GMCP-Liposomes

For the last dose in the above in vivo antitumor efficacy study, the mice in CP-liposomes (15 mg/kg) group and GMCP-liposomes group were, respectively, intravenously injected with DiR-labeled CP-liposomes and DiR labelled GMCP-liposomes instead. A total of 6 mice in each group were sacrificed at 18th hour post dose and tumors and the major organs (heart, liver, lung, spleen, kidney, and brain) were dissected for near-infrared imaging using an IVIS Living Image software, version 4.4 (Caliper Life Sciences, Hopkinton, MA, USA).

In order to observe the dynamic biodistribution of DiR-labelled liposomes, six 4T1 tumor-bearing mice (tumor volume of ~1000 cm^3^) were injected through the tail vein with DiR-labeled CP-liposomes (3 mice) and DiR labelled GMCP-liposomes (3 mice), respectively. They were whole-body imaged at the 0.5th, 1st, 2nd, 4th, 6th, 8th, 12th, 24th, 48th, 72nd, and 96th post dose using the IVIS Living Image software (version 4.4, Caliper Life Sciences, Hopkinton, MA, USA).

### 2.15. Statistical Analysis

Statistical analysis of the experimental data was performed using the Statistical Package for the Social Sciences software, and IC50 values were calculated by GraphPad Prism software, version 6.01 (GraphPad Software, La Jolla, CA, USA). In vitro and in vivo results were analyzed by t-test and one-way analysis of variance. The value of *p* < 0.05 was considered statistically significant.

## 3. Results and Discussion

### 3.1. Preparation and Characterization of CP-Liposomes and GMCP-Liposomes

In the preliminary experiment, both the film-sonication method and ethanol injection method were tried to prepare CP-liposomes, and the former produced smaller liposomes with more narrow distribution, thus it was adopted to prepare all the liposomes needed in the subsequent study. Meanwhile, 4:1:1 was chosen as the optimal feeding ratio of SPC, PPD and CBD. As shown in Table 1, when using PPD as the membrane stabilizer instead of cholesterol, the liposomes were easier to prepare and were a smaller size. The increase in size indicated that Mal was successfully inserted into the lipid bilayer through the long carbon chain. The obtained CP-liposome exhibited a mean particle size of 138.8 nm with a PDI value of 0.245, while GMCP-liposome exhibited a little larger mean particle size of 179.3 nm with similar particle size distributions (PDI value, 0.267) (Appendix A and Figure 3). TEM observation revealed a spherical morphology of CP-liposome with obvious liposomal bilayer. GP liposomes also presented as a spheroidal morphology, but no liposomal bilayer was observed, probably due to the interference of the anchored maltosyl residues. The particle sizes of the CP-liposome and GMCP-liposome measured by TEM were smaller than those of measured by DLS, as often reported. The Drug-loaded amount of CBD was 14.26% for CP-liposomes and 12.17% for GMCP-liposomes, both with a nearly 100% encapsulation efficiency.

### 3.2. Stability of CP-Liposome and GMCP-Liposome

Both CP-liposome and GMCP-liposome are quite stable in various physiological media. During the 12 h of incubation at 37 °C, the particle size (Figure 4a,b) and PDI values (Appendix A) of the CP-liposome and GMCP-liposome were nearly unchanged in normal saline, 5% glucose solution, PBS (pH = 7.4), simulated gastric fluid (SGF), simulated intestinal fluid (SIF) and plasma, and no aggregation was observed, indicating good suitability for intravenous injection and oral administration.

### 3.3. X-ray Diffraction Investigation and Differential Scanning Calorimetry

It was clear from the XRD patterns (Figure 4c) that free CBD, PPD and Mal all displayed sharp diffraction peaks, indicating their existence in a crystalline form. However, no diffraction peak was observed in lyophilized CP-liposomes, and GMCP-liposomes, indicating the absence of CBD or PPD crystalline in the resultant liposomes. The very weak CBD or PPD diffraction peaks in the physical mixture corresponding to CP-liposomes and GMCP-liposomes was attributed to the low melt-point of SPC (below 0 °C), which led to CBD-SPC interaction or PPD- SPC interaction during the process of well mixing.

A similar phenomena also occurred in the DSC pattern (Figure 4d), where CBD displayed a sharp peak at 68.03 °C; corresponding to its melting point of CBD, PPD displayed a weak peak and these two peaks totally displayed in lyophilized CP-liposomes and GMCP-liposomes, demonstrating that there was an interaction between both CBD-excipients and PPD-excipients.

### 3.4. The HPLC Standard Curve and Drug Release Profile

The standard curve of CBD was obtained by HPLC analysis, the formula was y = 0.3662x + 0.3981 with a liner range of 0.5–100 μg/mL and R^2^ being 0.999. CP-liposomes and GMCP-liposomes displayed quite similar biphasic in vitro drug release behavior, a relatively quick release phase within 12 h with a cumulative drug release reaching approximately 50% (Figure 4e,f), followed by a sustained release phase up to 90% at the 144th hour (Figure 4e). The sustained and prolonged drug release may help to reduce the drug leakage of CP-liposomes and GMCP-liposome during blood circulation and benefit drug accumulation in tumors.

### 3.5. In Vitro Antitumor Cell Growth Inhibition

The in vitro antitumor activity of liposomal CBD was examined against 4T1, MCF-7, A549, Hela, HepG2 and C6 cell lines using free CBD as a control were carried out in order to select the most sensitive one towards CP-liposome and GMCP-liposome (Appendix A). The different cell lines were treated with CP-liposome, GMCP-liposome, and free CBD at different concentrations for 72 h. The results indicated that liposomal CBD, in combination with an equal amount of PPD, exhibited a much stronger antitumor cell growth inhibition than free CBD at all concentrations and on all the six tested tumor cell lines, among which the 4T1 cell line was most sensitive to CBD. Free CBD showed an IC50 of 5.44 µg/mL, while CP-liposomes and GMCP-liposomes showed much stronger (32-fold and 17-fold, respectively) in vitro antitumor potency (IC50, 0.15 µg/mL and 0.28 µg/mL, respectively) (Table 1). There was no significant difference in IC_50_ value between CP-liposomes and GMCP-liposomes in all test cell lines. IC50 values of CBD-liposomes (PC:chol:CBD = 4:1:1) and PPD-liposomes (PC:PPD = 4:1) were 0.2232 and 0.2382 μg/mL, while the IC50 value of CP-liposome is 0.1486, which is lower than that of CBD-liposomes or PPD-liposomes (Appendix A). This indicated that the CBD liposome alone had a growth inhibitory effect on the 4T1 cell line, but it is not as effective as using it together with PPD. Taken together, CP-liposome could be used as a promising drug delivery system with their extremely high-efficiency in vitro therapeutic effects. Although treatment of the GMCP-liposome in vitro may not have had a significant effect, it might show potential targeting ability in vivo.

### 3.6. In Vivo Antitumor Efficacy

The BALB/c mice bearing 4T1 cells were chosen in our study to assess whether liposomal encapsulation in addition to combined therapy of PPD could effectively improve the therapeutic efficacy of CBD and whether the further modification of maltose residue on the surface of liposomes could lead to further improvement in the in vivo tumor growth inhibition. When the mean tumor volume of the normal saline group reached 2000 mm^3^, the experiment was terminated, and all the mice were sacrificed according to the ethical requirements of laboratory animals.

It was clearly seen from the tumor volume change curve (Figure 5a) and the tumor inhibition rate (TIR) calculated on the basis of tumor weight (Table 2) that CP-liposomes demonstrated very good dose-dependent anti-tumor efficacy, with a TIR of 46.0%, 67.4%, 82.2% for 5, 15, 45 mg/kg, respectively. In contrast, PTX injection (8 mg/kg) resulted in a TIR of only 64.4%. This demonstrated that the CBD-PPD combined therapy, together with liposomal co-encapsulation could be a very promising antitumor strategy. At the same a dose of 15 mg/kg, CBD-liposomes, and PPD-liposomes (15 mg/kg of PPD) showed a TIR of 46.8% and 50.8%, significantly lower than that of CP-liposomes (TIR, 67.4%, *p* < 0.05), suggesting the presence of a synergistic effect between CBD and PPD. However, further glycosyl-modification on the surface of liposomes failed to further improve the antitumor therapeutic efficacy (TIR, 71.0% for GMCP-liposomes). At the same dose of 45 mg/kg, orally administrated CP-liposomes only led to a TIR of 56.8%, while intravenously injected CP-liposomes achieved a high TIR of 82.2%; this may be due to the fact that the intravenously injected nanomedicine can much better benefit from the effect of EPR than orally administrated nanomedicine. Meanwhile, the latter may also suffer from insufficient oral bioavailability.

Although there was no significant bodyweight loss for mice in all the groups and no difference was observed in the body weight change among these groups (Figure 5b). All the mice were vigorous and behaved normally, except those mice in the PTX injection group that partly curled up with less movement. This indicated that the mice had a good tolerance to CBD liposomes.

Furthermore, the liver and spleen indices of the test groups showed no significant difference compared to that of the normal saline group (Table 2), indicating the low systemic toxicity and good tolerance of mice to the liposomal CBD.

### 3.7. The In Vivo Biodistribution

In order to investigate whether glycosylated modification can improve tumor accumulation. Another two groups of 4T1 tumor-bearing mice (three mice in each group, tumor volume about 1000 mm^3^) were intravenously injected with DiR-labled CP-liposome and DiR-labled GMCP-liposome at the dose of 15 mg/kg equivalent CBD. As illustrated in Figure 6a, CP-liposomes were mainly located in the liver throughout the whole observation period of 96 h, and no obvious accumulation in tumor was noticed, probably due to their failure to achieve long circulation and then quick uptake by the liver mononuclear phagocyte system. In case of GMCP-liposomes, although the main distribution was found in liver within 2 h post dose, significant tumor accumulation was observed after the 2nd hour, then, gradually strengthened, reached a plateau at the 24th hour, and then maintained a high accumulation level till the end of the dynamic observation at the 96th hour, demonstrating that the glycosylated modification did greatly improve the tumor targetability of liposomal CBD (Figure 6b).

However, in the in vivo antitumor efficacy study, GMCP-liposomes failed to achieve significant improvement in therapeutic efficacy in comparison with CP-liposomes (TIR, 71.0 ± 14.4% vs. 67.4 ± 6.9, *p* > 0.05). Luckily for the last dose in the in vivo antitumor efficacy study, the mice in the CP-liposomes (15 mg/kg) group and GMCP-liposomes group were intravenously injected with DiR-labelled CP-liposomes and GMCP-liposomes to assess their biodistribution after multiple doses. This time, no dynamic imaging was performed; all the mice were sacrificed at the 18th hour post dose, then the tumors and the dissected major organs were fluorescently imaged. CP-liposomes and GMCP-liposomes displayed quite similar organ bio-distribution (Figure 6c–f), mainly in the liver and spleen, then in the tumor and in the lungs. Additionally, the ratio of tumor fluorescence/liver fluorescence was calculated to be 0.71 for CP-liposomes and 0.75 for GMCP-liposomes. This result was different from that showed in the dynamic observation. Such a deviation was seldom reported, and we still do not know the exact reason. A more elaborate design and direct comparison need to be performed, especially with a focus on the difference resulting from multiple dosing and single dosing; the difference resulted from the dynamic observation of whole-body imaging and direct fluorescent imaging of the dissected organs.

## 4. Conclusions

In order to develop an effective and well-tolerated anti-tumor agent, CBD, a compound showing wide physiological activities, was proposed to combine with PPD, another compound derived from well-known herbal medicine, Ginseng, for tumor treatment. The combined therapy based on these two low toxic natural compounds turned out to have a very good therapeutic efficacy in 4T1 graft tumor, a very aggressive and metastatic tumor. Neither CBD nor PPD proved to be a potent anti-tumor agent; however, when co-encapsulated into liposomes, the combined therapy achieved a high tumor inhibition rate of 82.2%, much higher than 8 mg/kg of PTX injection (64.4%). It is noteworthy that no side effects were observed, and all the mice behaved normally and vigorously. This result provides a preliminary foundation for the development of novel anti-tumor agents based on CBD, characteristic of good effectiveness and safety.

There was no pegylation performed for the liposomes used in this study, so the pegylation can probably further enhance the in vivo antitumor efficacy of CP-liposomes due to the long circulation and the resultant better EPR effect.

Ginsenosides Rg3 and Rh2, the glucosides of PPD, which also showed antitumor activity and many health beneficial effects, may also achieve a synergetic effect with CBD. It has been demonstrated that PTX co-loaded Rg3 or Rh2 liposomes realized very excellent in vivo antitumor efficacy and achieved a “one stone four birds” effect. Therefore, CBD promises to exert better anti-tumor action in combination with Ginsenosides Rg3 and Rh2. Further experiments regarding this are ongoing in our laboratory.

Cancer cachexia (CCA) is a multifactorial and wasting symptom commonly seen in cancer patients. CCA attacks about 50–80% of cancer patients [60] and leads to 25% of cancer deaths [61]. It has been proved that cannabinoid has considerable potential to improve the appetite, body weight, body fat level, caloric intake, mood and quality of life in patients of these kind of diseases [37]. Since CBD plus PPD themselves have direct and potent antitumor action, CP-liposomes may exert expectant positive effects in cancer cachexia improvement, alone or in combination with other antitumor strategies. Therefore, it can be assumed that CP-liposomes might also be considered a potential candidate for neoadjuvant and/or adjuvant interventions in oncology.

## Figures and Tables

**Figure 1 pharmaceutics-14-01533-f001:**
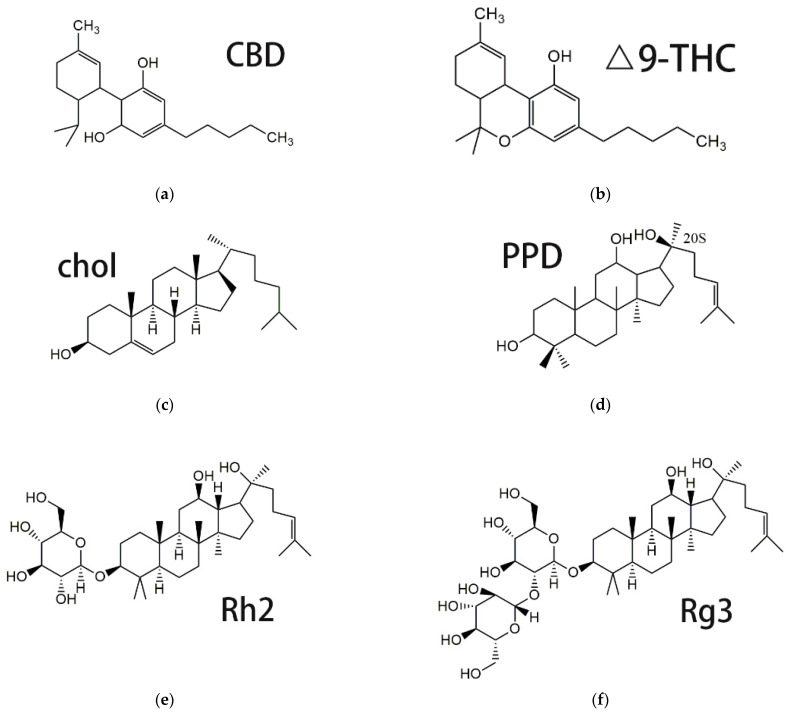
Chemical structure of CBD (**a**), Δ^9^-THC (**b**), Chol (**c**), PPD (**d**) and ginsenoside Rh2 (**e**) and Rg3 (**f**).

**Figure 2 pharmaceutics-14-01533-f002:**
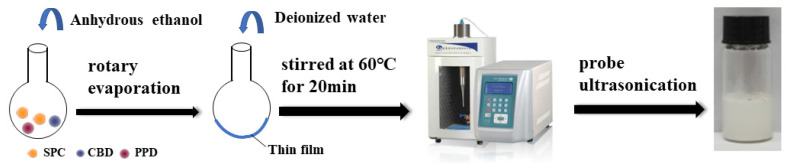
The preparation of CP-liposomes and GMCP-liposomes.

**Figure 3 pharmaceutics-14-01533-f003:**
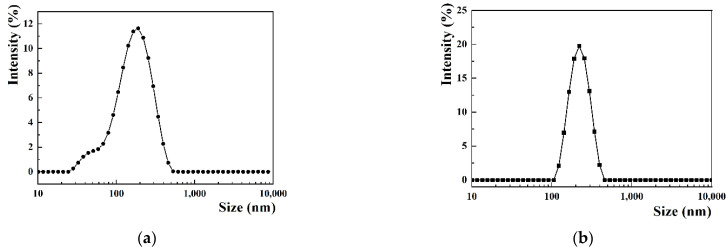
Particle size distribution of CP-liposome (**a**) and GMCP-liposome (**b**), and transmission electron microscopy images of CP-liposome (**c**) and GMCP-liposome (**d**).

**Figure 4 pharmaceutics-14-01533-f004:**
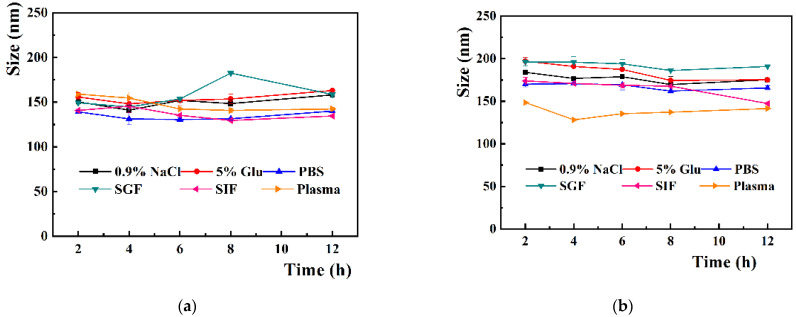
The particle size change of CP-liposome (**a**) and GMCP-liposome (**b**) in various physical media at 37 °C, and the XRD (**c**) and DSC (**d**) pattern of lyophilized CP-liposome and GMCP-liposome in contrast to free CBD, free PPD, SPC, Mal, and the corresponding physical mixture. The accumulative in vitro CBD release from CP-liposomes and GMCP liposomes within 144 h (**e**) or within 24 h (**f**).

**Figure 5 pharmaceutics-14-01533-f005:**
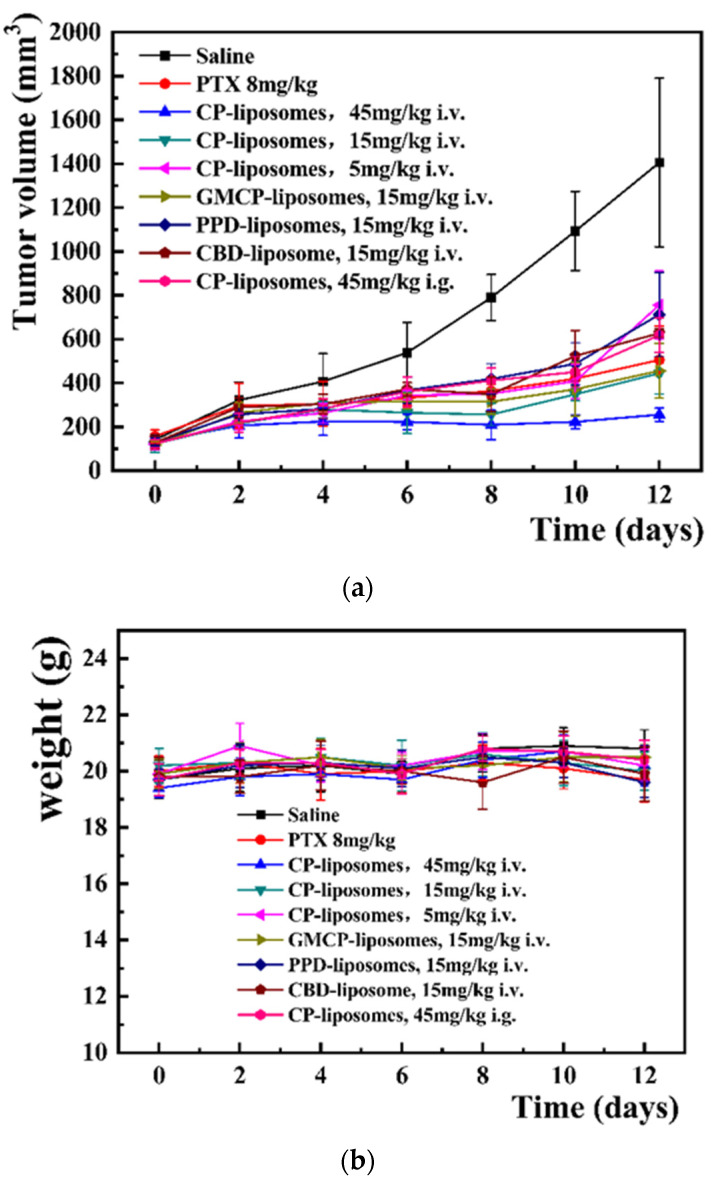
The in vivo anti-tumor efficacy of CBD-liposomes, PPD liposomes, CP-liposomes, GMCP-liposomes. (**a**) for tumor volume change curves, (**b**) for body weight change curve and (**c**) for the actual tumor image collected at the end of the experiment.

**Figure 6 pharmaceutics-14-01533-f006:**
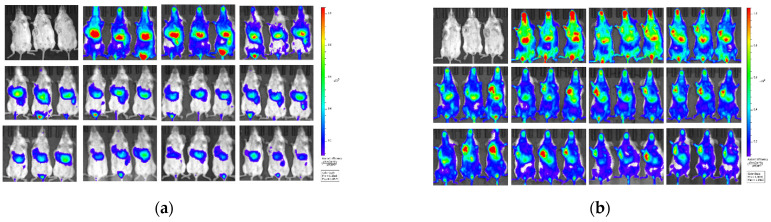
The in vivo biodistribution of DiR-labelled CP-liposome and Dir-labelled GMCP-liposome in 4T1 tumor-bearing mice. Dynamic in vivo bio-distribution of Dir-labelled CP-liposome (**a**) and DiR-labelled GMCP-liposome (**b**) at different time points (from left to right: 0 h, 0.5 h, 1 h, 2 h, 4h, 6 h, 8 h, 12 h, 24 h, 48 h, 72 h and 96 h post dose through tail vein. The fluorescent images of tumor and major organs of mice receiving DiR-labelled CP-liposomes (**c**) and DiR-labelled GMCP-liposomes (**d**) 16 h post dose (from left to right: tumor, heart, liver, spleen, lung, kidney, and brain) (n = 6, mean ± SD) and their Fluorescence semi-quantitative analysis (**e** for CP-liposomes, **f** for GMCP-liposomes).

**Table 1 pharmaceutics-14-01533-t001:** IC_50_ values (μg/)mL of free CBD, CP-liposome, and GMCP-liposome against different tumor cell lines after 72 h incubation (n = 5).

	4T1	MCF-7	A549	C6	Hela	HepG2
Free CBD	5.44	23.94	13.18	7.14	20.57	20.57
CP-liposome	0.1486	2.793	6.036	4.014	4.750	6.490
GMCP-liposome	0.2750	3.699	5.977	3.303	5.824	5.867

**Table 2 pharmaceutics-14-01533-t002:** In vivo antitumor efficacy and safety evaluation.

Group	Tumor Weight (g)	Tumor Inhibition Rate (%)	Liver Index	Spleen Index
Saline	0.83 ± 0.19		0.0589 ± 0.0077	0.0227 ± 0.0067
PTX injection (8 mg/kg)	0.30 ± 0.12 *	64.4 ± 14.3	0.0506 ± 0.0066	0.0164 ± 0.0023
CP-liposomes, 45 mg/kg i.v.	0.15 ± 0.05 *	82.2 ± 5.6	0.0517 ± 0.0081	0.0160 ± 0.0025
CP-liposomes, 15 mg/kg i.v.	0.27 ± 0.06 *	67.4 ± 6.9	0.0547 ± 0.0033	0.0187 ± 0.0032
CP-liposomes, 5 mg/kg i.v.	0.45 ± 0.17 *^$&^	46.0 ± 20.3	0.0589 ± 0.0038	0.0183 ± 0.0049
GMCP-liposomes, 15 mg/kg i.v.	0.24 ± 0.12 *	71.0 ± 14.4	0.0556 ± 0.0049	0.0196 ± 0.0067
PPD-liposomes, 15 mg/kg i.v.	0.43 ± 0.26 *^$^	46.8 ± 12.0	0.0529 ± 0.0046	0.0211 ± 0.0035
CBD-liposome, 15 mg/kg i.v.	0.41 ± 0.12 *^#^	50.8 ± 15.7	0.0526 ± 0.0038	0.0203 ± 0.0030
CP-liposomes, 45 mg/kg i.g.	0.36 ± 0.08 *^#^	56.8 ± 9.2	0.0523 ± 0.0035	0.0214 ± 0.0047

The results are presented as the mean ± SD, n = 6. * *p* < 0.01 vs. normal saline. ^#^
*p* < 0.05 vs. CP-liposomes 45 mg/kg i.v. ^$^
*p* < 0.01 vs. CP-liposomes 45 mg/kg i.v. ^&^
*p* < 0.05 vs. GMCP-liposomes 15 mg/kg i.v.

## Data Availability

Not applicable.

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
