# Peer review of "Improved Therapeutic Efficacy of CBD with Good Tolerance in the Treatment of Breast Cancer through Nanoencapsulation and in Combination with 20(S)-Protopanaxadiol (PPD)"

_pharmaceutics, 2022, doi:10.3390/pharmaceutics14081533_

Round 1

Reviewer 1 Report

This manuscript showed the effect of CBD with nanoencapsulation as a breast cancer therapy.
The authors performed the capsulation of CBD and the treatment studies in vivo.
Overall, experiments are well conducted.
CBD and its treatment strategy is interested to the breast cancer therapy.
Addressing the comments would be strengthen the conclusion.

Comments
1. The authors should discuss the mechanism and target genes of CBD. 4T1 cells are aggressive breast cancer cell lines; therefore, the mechanism is important and interesting for the further studies.
Additionally, the effect in other types of cancer should be discussed.

2. Liposomes has the property which concentrated to liver and adipose tissues. Toxicity of CBD capsule in the liver should be discussed in more detail.

3. The need of liposome capsulation should be discussed. The reviewer think that single-CBD treatment would be acceptable. 

Reviewer 2 Report

Summary/General Comments:

In the manuscript by Fu et al., the authors presented novel data examining the anti-cancer effects of cannabidiol (CBD) in combination with 20(S)-protopanaxadiol co-loaded into liposomes. The authors showed that CP-liposomes and GMCP-liposomes decreased the IC50 values, as compared to free CBD, in several breast, lung, glioma, cervical, and liver cancer cell lines. Although not significantly, GMCP-liposomes also decreased tumor growth in 4T1-induced tumors in mice. Despite the value of the subject matter, the manuscript would be significantly strengthened after text and figure editing, as well as addressing the comments below.

Comments:

·        Significant language proofing/editing is needed to improve the overall clarity and accuracy of the manuscript.

o   Numerous run on/lengthy and incomplete sentences

o   Line 12: Italicize Cannabis sativa

o   Line 22: “4T1 breast tumor-bearing mouse model…”

o Line 24: Did glycosylation modification fail to significantly increase/decrease/alter the TIR of CP-liposomes?

o   Line 40: Remove last “in”

o   Line 51: “has been applied in the treatment of a wide range of neurodegenerative diseases”

o   Define “PTX”

o   Line 145: Proof heading

o   Line 267: “0.245nm”

o   Line 327: Remove period after “control”

 Results and Figures:

o   Figure 1: It would improve clarity in the text if the individual panels in Figure 1 were labeled a-f so that the specific panel/chemical structure could be referenced in the text.

o   Figure 2 should be improved to more clearly depict the methodology

o   Figure 3:

§  What is the rationale for selecting 15mg/kg dosage of GMCP-liposomes, PPD-liposomes, and CBD-liposomes?  5 and 45 mg/kg of CP-liposomes was also tested.

§  What is the amount of each drug found in the blood after intravenous administration and oral administration of CP-liposomes? Are they similar?

o   Lines 273-275: Is this data presented in a figure? If not, state “data not shown”.

o   Figure 5 is not referenced in the text.

§  Define SIF and SGF.

§  The inset graph in 5e is too small to clearly read, making it difficult to interpret the figure panel

o   Figure 7:

§  Provide the cell viability data for all other cell lines. Why were they not shown?

§  The in vitro MTT/IC50 data do not show the effect of the treatments on tumor growth. Consider revising to tumor/cancer cell growth inhibition.

§  Further control experiments should be done to see if the liposome alone has a growth inhibitory effect on the cell lines.

§  By what mechanism do the CP-liposomes and GMCP-liposomes inhibit cell viability? Is it the same mechanism as “Free CBD”? Additional experiments should be performed to answer these questions.

o   Figures 8-9: The figure text is very small, making it difficult to interpret the data presented in these results.

o   Figure 9a/b: Images at 0h should be provided to establish a baseline.

o   Figure e/f: The presentation of the Tumor/Liver ratio is distracting in these graphs and should be removed. 

·       Discussion/Conclusions:

o   This section needs to discuss the implications and impact of this work in a broader scope. 

·       References:

o   Additional references should be cited to support the growth inhibitory effects of CBD on leukemia, lung cancer, cervical cancer, prostate cancer, and melanoma (Lines 52-53)

o   In-text citations need to be reformatted in line 59

o   References should be cited for the information in Lines 63-71

·       Additional note:

o   The authors mentioned in the abstract that the incorporation of ‘Mal’ into the GMCP-liposomes “acts as a ligand targeting GLUT1 receptor high expressed on tumor cells.” However, this was not shown or discussed in the rest of the manuscript. This should be addressed in the manuscript or removed from the abstract.

Reviewer 3 Report

The article is interesting but lacks mechanism of action studies. The compound described in the study needs further exploration.

Round 2

Reviewer 3 Report

The article is interesting, However the mechanism by which the Cannabidiol can reduce the tumor tumor growth is not clear in the article. The MOA studies are missing in the article.
